# Carboxyhemoglobinemia in Critically Ill Coronavirus Disease 2019 Patients

**DOI:** 10.3390/jcm10122731

**Published:** 2021-06-21

**Authors:** Hina Faisal, Syeda T. Ali, Jiaqiong Xu, Tariq Nisar, Mahmoud Sabawi, Eric Salazar, Faisal N. Masud

**Affiliations:** 1Department of Surgery, Houston Methodist Hospital, Houston, TX 77030, USA; tooba.najeeb@gmail.com; 2Center for Outcomes Research, Houston Methodist Research Institute, Houston, TX 77030, USA; SXu@houstonmethodist.org (J.X.); tnisar@houstonmethodist.org (T.N.); 3Department of Pharmacy, Houston Methodist Hospital, Houston, TX 77030, USA; msabawi@houstonmethodist.org; 4Department of Pathology and Genomic Medicine, Houston Methodist Hospital, Houston, TX 77030, USA; esalazar@houstonmethodist.org; 5Department of Anesthesia & Critical Care Medicine, Houston Methodist Hospital, Houston, TX 77030, USA; fmasud@houstonmethodist.org

**Keywords:** coronavirus disease 2019, carboxyhemoglobinemia, carbon monoxide, incidence, etiology, elevated carboxyhemoglobin

## Abstract

Carboxyhemoglobinemia is a common but a serious disorder, defined as an increase in carboxyhemoglobin level. Unfortunately, there are few data on carboxyhemoglobinemia in coronavirus disease 2019 (COVID-19) patients. Therefore, our study aimed to evaluate the incidence and etiologies of carboxyhemoglobinemia in COVID-19 patients and determine any association between carboxyhemoglobinemia and novel coronavirus infection. A retrospective chart review was performed at an academic medical center for all inpatient COVID-19 cases with either single or serial carboxyhemoglobin (COHb) levels from March 2020 through August 2020.Our study demonstrates that carboxyhemoglobinemia in COVID-19 patients is due to sepsis, hemolysis, and cytokine storm, triggered by the novel coronavirus infection sequela and is not directly from the virulence of novel coronavirus. Given the coexisting illnesses in critically ill COVID-19 patients, it is impossible to establish if coronavirus virulence was the culprit of elevated COHb levels. Moreover, our study found a high incidence of carboxyhemoglobinemia in critically ill COVID-19 patients. The oxygen saturation measured by pulse oximetry can be inaccurate and unreliable; however, our study could not demonstrate any uniform results on the discrepancy between oxygen saturation measured by pulse oximetry and arterial blood gas. In this study, COHb levels were measured using a CO-oximeter. Therefore, we recommend monitoring the COHb level routinely in critically ill COVID-19 patients to allow more effective and prompt treatment.

## 1. Introduction

Coronavirus disease 2019 (COVID-19) from a novel human coronavirus (severe acute respiratory syndrome coronavirus 2, SARS-CoV-2) emerged to become a pandemic and has shaken the world and left medical experts searching for answers. One such question concerns the increased incidence of carboxyhemoglobinemia observed in COVID-19 patients. Since the pandemic is ongoing, accurate epidemiological characteristics are yet to be determined. However, the case fatality rate with the novel coronavirus is 2–3%, which is alarming enough to thoroughly study its morbidity and mortality [1]. The majority of SARS-CoV-2 infected patients are mildly symptomatic or asymptomatic. However, severe COVID-19 can lead to a more serious lower respiratory infection requiring hospitalization and intensive care unit admission due to pneumonia with hypoxemia, acute respiratory distress syndrome, severe systemic inflammation, and cytokine release syndrome [2,3,4].

Carboxyhemoglobinemia is defined as an increase in COHb level, which originates from the oxidative degradation of hemoglobin by the enzyme heme oxygenase, resulting in reducing its oxygen-carrying capacity [5]. Elevated COHb level is common in sepsis [6], hemolysis [7], and severe inflammatory conditions [8], and can cause profound hypoxia and, ultimately, lead to neurocognitive deficits and myocardial depression [9].

While there are few reports to date on elevated COHb levels in COVID-19 patients [10,11] at our institute, we observed high COHb levels in critically ill COVID-19 patients, which our physician checked due to discrepancies between oxygen saturation measured by pulse oximetry and arterial blood gases. Therefore, we hypothesize that raised levels of both endogenous carbonmonoxide and elevated COHb levels may continue to increase to toxic levels in critically ill COVID-19 patients as overwhelmed and failed organs and systems lose their ability to absorb, metabolize, and excrete toxins.

Our study aimed to evaluate the incidence and etiologies of carboxyhemoglobinemia in COVID-19 patients and determine any association between carboxyhemoglobinemia and novel coronavirus infection, whether elevated COHb level is due to coronavirus infection or by the medications or/and drugs used for the treatment of COVID-19 disease.

## 2. Methods

A retrospective chart review was performed at an academic medical center for all inpatient COVID-19 cases with randomly checked either single or serial COHb levels from March 2020 through August 2020. Study subjects consisted of 431 adult patients > 18 years, admitted to our institution with a confirmed laboratory diagnosis of SARS-CoV-2 infection or COVID-19. The patients with a previous history of congenital hemoglobinopathies and a history of ingestion/exposure to toxins at the time of admission (e.g., carbon monoxide poisoning, ingestion of pesticide, and insecticides) were excluded. The study proposal received Houston Methodist Research Institute (HMRI) institutional review board (IRB) approval (ID # PRO00026061). The centralized medical center billing records were queried for the volume of total inpatient encounters diagnosed with SARS-CoV- 2 Infections/COVID-19, followed by calculating the overall incidence of carboxyhemoglobinemia. At our institute, RadiometerABL80 FLEX CO-OX blood gas analyzers are utilized to measure COHb levels. In addition, the analyzers also provide the value of blood gas, electrolytes, and glucose.

Clinically significant COHb level or carboxyhemoglobinemia was defined as blood COHb concentrations of 2.0% or above in nonsmokers or former smokers and >5% in current smokers. Hypoxia was defined as oxygen saturation measured by pulse oximetry (SpO_2_) < 94%. Four hundred thirty-one patients were randomly assessed for COHb levels at any point during their disease course.

Standardized data were collected on demographic features:smoking history, the presence of prior comorbidities (obesity, diabetes, hypertension, chronic pulmonary disease, renal and electrolyte disorders, anemias, and other heart and vascular diseases), sepsis, and medications used during hospital admission.Laboratory data included hemoglobin, hematocrit, lactate dehydrogenase (LDH), and total bilirubin, and COHb level. Elevated LDH and total bilirubin were defined as LDH > 225 U/L and total bilirubin > 1.2 mg/dL, respectively. Low hemoglobin and hematocrit were defined as venous hemoglobin < 14 g/dL for males and <12 g/dL for females, and venous hematocrit<40% for males and <36% for females.

## 3. Statistical Analysis

Baseline characteristics and clinical risk factors were summarized according to carboxyhemoglobinemia status. All data were presented as mean±SD for continuous variables and number (%) for categorical variables. Chi-square or Fisher’s exact test for categorical variables and t-test or Mann–Whitney test for continuous variables were used to compare patients between normal and abnormal carboxyhemoglobinemia. The Shapiro–Wilk test was used to test normality assumption. Univariable and multivariable logistic regression models or penalized maximum likelihood logistic regression models were used to examine risk factor associations with carboxyhemoglobinemia, as appropriate. All analyses were performed with STATA version 16 (StataCorp. 2019. Stata Statistical Software: Release 16. College Station, TX, USA: StataCorp, LLC). Statistical significance was defined as two-tailed *p* < 0.05 for all tests.

## 4. Results

Among the entire cohort of 5708 patients with COVID-19 admitted to Houston Methodist Hospital System, only 431 patients had single or multiple measurements of COHb levels. One hundred fifteenpatients had elevated COHb levels, for an incidence of 26.7% [115/431] (95% CI, 22.6–31.1%). The mean initial COHb was 1.51 ± 1.22%. There was a decreasing trend with COHb levels in an abnormal or elevated COHb group throughout measurements (*p*-trend = 0.01). However, the trend in the COHb level remained the same for the normal COHb group (*p*-trend > 0.05) (Figure 1).

As shown in the Figure 1, 244 out of 431 (56.6%) patients had one blood analysis, 187 (43.4%) had more than one blood analysis. Thus, the maximum repetition of COHb level measurements was 65, and most patients had COHb levels checked on a median hospital day of 2.5 days {interquartile range (IQR): 0.1–7.6}.

Out of 431 patients, 215 (49.88%) patients were admitted to intensive care units or intermedicate care units. One hundred forty-nine out of 431 (34.57%) patients required intubation and mechanical ventilation. For the intubation medications, 137(91%) patients received propofol and rocuronium, and 12 (8.05%) patients received propofol and succinylcholine for rapid sequence intubation. The baseline characteristics of the patients through their first encounter admissions are presented in Table 1. There were no significant differences in age, gender, or BMI between abnormal and normal carboxyhemoglobin levels. Smoking status was statistically significant between abnormal and normal carboxyhemoglobin levels (*p* < 0.001). We found 56 (48.7%) nonsmokers, 45 (39.13%) former smokers, 14 (12.17%) with unknown status, and no active/current smokers in a total of 115 carboxyhemoglobinemia cases. Elevated carboxyhemoglobin level was also associated with pre-existing anemias compared to patients with normal carboxyhemoglobin levels (6.96% vs. 2.53%, *p* = 0.043). Many patients with elevated COHb levels were intubated and admitted to the intensive care and intermediate care units (*p* < 0.001) (Table 1).

Concerning risk factors for carboxyhemoglobinemia (Table 2), sepsis, hydroxychloroquine, tocilizumab, and antibiotics all had a statistically significant association with elevated COHb level (all *p* < 0.001). In addition, low hematocrit, elevated total bilirubin, and elevated LDH werealso statistically significant in association with elevated COHb level (0.003, 0.001, and 0.002, respectively).

On the univariable logistic regression model, COVID-19 patients with elevated COHb levels showed statistically significant association with former smokers; pre-existing anemias; sepsis; elevated total bilirubin; elevated LDH; and use of hydroxychloroquine, tocilizumab, and antibiotics (all *p* < 0.05). However, the multivariable analysis showed that pre-existing anemias, tocilizumab, antibiotics, and elevated total bilirubin were statistically significant and independently associated with elevated COHb levels (all *p* < 0.05, Table 3).

## 5. Discussion

Carboxyhemoglobinemia is a common and serious disorder. However, most published data have been limited to case reports and small case series [11], making the assessment of prevalence and risk factors difficult. Due to the scarcity of these data on a potentially life-threatening disorder and in light of the current pandemic, our study aimed to determine whether the novel coronavirus has any significant role in the pathogenesis of carboxyhemoglobinemia in COVID-19 disease. 

Our definition of carboxyhemoglobinemia was based on the fact that concentrations below the levels considered in our study are rarely associated with illness, even in critically ill and compromised patients [12,13].

In our study, the incidence of carboxyhemoglobinemia was 26.7%. To date, there are no prior published studies on the incidence of elevated COHb in COVID-19 patients. 

Causes of carboxyhemoglobinemia include carbon monoxide sources such as motor vehicles, propane stoves, and charcoal grills in poor ventilation [14]; and exposure to methylene chloride, usually by inhalation [15]. Drug-induced carboxyhemoglobinemia is rare but has been associated with high doses of sodium nitroprusside or can result from drug-induced hemolytic anemia [7,16,17]. Of the drugs that patients received in this study, beta-lactam antibiotics are associated with hemolytic anemia and can lead to carboxyhemoglobinemia [18]. In this study, we found a strong association between carboxyhemoglobinemia and the use of hydroxychloroquine, tocilizumab, and antibiotics. However, we cannot establish any direct association between elevated COHb and the use of these medications, which are commonly used in the management plan of critically ill COVD-19 patients.

Carboxyhemoglobinemia can also occur with any condition associated with increased heme catabolism [19,20]. The severity of hemolysis correlates closely with CO production and, in turn, measured COHb level [21]. In general, COHb increases due to hemolysis are at the level of 2–3% but can be higher [20,21]. Moreover, it can be a feature of severe inflammatory diseases, e.g., sepsis and pneumonia, due to increased expression of heme oxygenase induced by inflammatory cytokine [19]. In our study, most patients showed evidence of hemolysis (elevated LDH, total bilirubin, and low hematocrit); however, a causal association between hemolysis and COVID-19 disease could not be established as it may be secondary to other underlying conditions such as sepsis in a critically ill patient.

Cytokine release syndrome (CRS) is prevalent in COVID-19 patients. The inflammatory storm and subsequent immunopathologic changes in these patients affect the lungs and result in acute respiratory distress syndrome, which is the leading cause of death in COVID-19 patients [22]. Respiratory infections increase CO production in the lungs at least five-fold as a defense mechanism [23,24]. An increased expression and upregulation of hemeoxygenase in the liver, spleen, and lungs occurred under increased oxidative stress conditions, e.g., sepsis [24], with a notable increase in COHb concentration during the first three days of sepsis treatment [25].Similarly, in our study, a significant positive correlation was observed with most patients (75.65%) having underlying sepsis of varying severity, reinforcing the hypothesis of sepsis being the culprit for the development of carboxyhemoglobinemia.

Several studies hypothesize that an optimal level of COHb reduces mortality in critically ill medical and surgical patients [26]. Melley et al. showed that the minimum COHb level was significantly higher in patients who survived a short ICU stay (median 1.0 (0.9 to 2.8) days) following cardiothoracic surgery compared to patients who died in the ICU [27]. Our results do not confirm their observation; in fact, a marginally high level of COHb in critically ill COVID-19 patients was associated with an increased risk of mortality.

Apart from hyperbaric or normobaric oxygen therapy and supportive care, treatment options for carboxyhemoglobinemia are limited. However, there is interest in hydroxocobalamin and ascorbic acid as a potential therapy, given evidence for conversion of COHb to carbon dioxide in rat models [28]. Of note, hydroxocobalamin may result in falsely lower COHb levels [29].

A study by Roth et al. found an unexpected level of 28% COHb on a pulse CO-oximeter compared to false normal levels seen on a conventional pulse oximeter in the same patient; hence, pulse CO-oximeters are a reliable and noninvasive way of determining the levels of COHb and could be a valuable tool in COVID-19 patients presenting with a multitude of nonspecific symptoms [30].To summarize, this study demonstrates that carboxyhemoglobinemia in COVID-19 patients is due to sepsis, hemolysis, and cytokine storm, triggered by the novel coronavirus infection sequela and is not directly from the virulence of novel coronavirus. Furthermore, critically ill COVID-19 patients have several risk factors for developing this fatal condition, such as COVID-19 associated viral pneumonia, sepsis, and multisystem organ failure, which are themselves known for elevating COHb levels. Given the coexisting illnesses in critically ill COVID-19 patients, it is impossible to establish if coronavirus virulence was the culprit of elevated COHb levels.

This research has several limitations due to the inherent nature of a retrospective study. First, COHb levels were checked only in hypoxic patients, creating possible sample bias. Second, carbon monoxide competes with oxygen for the same binding sites on the hemoglobin molecule. Since supplemental oxygen and mechanical ventilation are common interventions in COVID-19 associated pneumonia that can contribute to increased lung elimination of carbon monoxide, the quality of the data may have been hampered by falsely decreasing COHb levels. Third, the laboratory markers for anemia assessment were not collected on the same day as the COHb levels, creating information bias; thus, hemolysis results need to be interpreted with caution in association with carboxyhemoglobinemia. Last, the lack of data on the number of mechanically ventilated patients could pose a selection bias. Therefore, more research needs to be done, excluding patients with sepsis and anemia, to see if COVID-19 infection is per se causing the COHb spike.

## 6. Conclusions

The incidence of carboxyhemoglobinemia is high in critically ill COVID-19 patients. However, given the coexisting illnesses in critically ill COVID-19 patients, it is impossible to establish if coronavirus virulence was the culprit of elevated COHb levels. In addition, the oxygen saturation measured by pulse oximetry can be inaccurate and unreliable in the case of elevated COHb levels in critically ill COVID-19 patients. Therefore, we recommend monitoring the COHb level routinely in critically ill COVID-19 patients to allow more effective and prompt treatment.

## Figures and Tables

**Figure 1 jcm-10-02731-f001:**
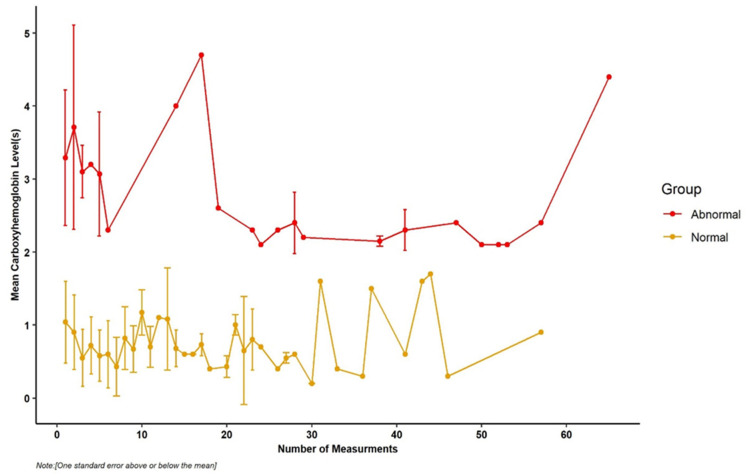
Time course on the mean carboxyhemoglobin level of each patient with elevated and normal carboxyhemoglobin levels.

**Table 1 jcm-10-02731-t001:** Baseline characteristics.

	Total	Carboxyhemoglobin		*p*-Value
		Normal	Abnormal	
	*n* = 431	*n* = 316	*n* = 115	
Ages	60.59 ± 15.77	60.19 ± 16.30	61.67 ± 14.24	0.39
**Sex (*n*, %)**				
Female	185 (42.92)	138 (43.67)	47 (40.87)	0.66
Male	246 (57.08)	178 (56.33)	68 (59.13)	0.66
BMI	32.38 ± 8.32	31.97 ± 8.10	33.51 ± 8.88	0.12
Obesity	206 (55.23)	150 (54.55)	56 (57.14)	0.72
**Smoking status (*n*, %)**				
Never	252 (58.47)	196 (62.03)	56 (48.70)	0.015
Former	130 (30.16)	85 (26.90)	45 (39.13)	0.018
Current	24 (5.57)	24 (7.59)	0 (0.00)	<0.001
Unknown	25 (5.80)	11 (3.48)	14 (12.17)	0.002
Diabetes Mellitus	50 (11.60)	42 (13.29)	8 (6.96)	0.088
Essential Hypertension	68 (15.78)	56 (17.72)	12 (10.43)	0.074
Heart and other vascular disease	36 (8.35)	29 (9.18)	7 (6.09)	0.43
Renal and electrolyte disorders	124 (28.77)	84 (26.58)	40 (34.78)	0.12
Chronic pulmonary disease	16 (3.71)	13 (4.11)	3 (2.61)	0.58
Pre-existing Anemias	16 (3.71)	8 (2.53)	8 (6.96)	0.043
Intensive care unit/ Intermediate care unit	215 (49.88)	122 (38.61)	93 (80.87)	<0.001
Intubation and Mechanical Ventilation	149 (34.57)	69 (21.84)	80 (69.57)	<0.001

Data are presented as mean ± SD for continuous variables and *n* (%) for categorical variables. Chi-square or Fisher’s exact test for categorical variables and *t*-test or Mann–Whitney test for continuous variables was used to compare patients between abnormal and normal carboxyhemoglobin level, as appropriate.

**Table 2 jcm-10-02731-t002:** Number (%) of Risk Factor by Carboxyhemoglobinemia.

	Total	Carboxyhemoglobin		*p*-Value
		Normal	Abnormal	
	*n* = 431	*n* = 316	*n* = 115	
Carboxyhemoglobin Level (Mean, SD)	1.51 ± 1.22	0.90 ± 0.54	3.19 ± 0.97	<0.001
Carboxyhemoglobin Level (Median, IQR)	1.10 (0.7–2.20)	0.8 (0.6–1.1)	3.0 (2.4–3.8)	<0.001
Sepsis	87 (20.19)	0 (0.00)	87 (75.65)	<0.001
Hydroxychloroquine	8 (1.86)	0 (0.00)	8 (6.96)	<0.001
Tociluzimab	57 (13.23)	0 (0.00)	57 (49.57)	<0.001
Antibiotics	96 (22.27)	0 (0.00)	96 (83.48)	<0.001
Low hematocrit	342 (79.35)	240 (75.95)	102 (88.70)	0.003
Low hemoglobin	375 (87.01)	269 (85.13)	106 (92.17)	0.074
Total bilirubin	*n* = 414	*n* = 299	*n* = 39	
Elevated total bilirubin	77 (18.60)	38 (12.71)	39 (33.91)	<0.001
Lactate dehydrogenase	*n* = 401	*n* = 288	*n* = 113	
Elevated Lactate dehydrogenase	368 (91.77)	257 (89.24)	111 (98.23)	0.002

Number (%) of risk factor by carboxyhemoglobinemia;data are presented as mean ± SD, median, and interquartile range (IQR) for continuous variables and *n* (%) for categorical variables. Chi-square or Fisher’s exact test for categorical variables and *t*-test or Mann–Whitney test for continuous variables was used to compare patients between abnormal and normal carboxyhemoglobin levels, as appropriate.

**Table 3 jcm-10-02731-t003:** Odds ratio (OR) and 95% CI of risk factors associated with carboxyhemoglobinemia from the logistic regression models.

	Univariable		Multivariable	
	OR (95% CI)	*p*-Value	OR (95% CI)	*p*-Value
Smoking status				
Never *	Reference		Reference	
Former	1.85 (1.16–2.95)	0.01	2.75 (0.97–7.75)	0.056
Current	0.07 (0–1.19)	0.066	0.43 (0.02–8.44)	0.58
Unknown	4.38 (1.91–10.04)	<0.001	0.73 (0.02–8.44)	0.86
Pre-existing anemias	2.88 (1.05–7.86)	0.039	7.13 (1.46–34.91)	0.015
Sepsis	1943 (117–32151)	<0.001	1.45 (0.03–76.16)	0.86
Hydroxychloroquine	50.05 (2.86–874)	0.007	0.22 (0–10.14)	0.44
Tociluzimab	622 (38–10208)	<0.001	130 (3.81–4457)	0.007
Antibiotics	3132 (187–52363)	<0.001	955 (20–46424)	0.001
Low hematocrit	2.48 (1.32–4.68)	0.005	5.0 (0.15–168.64)	0.37
Low hemoglobin	2.06 (0.97–4.35)	0.059	1.26 (0.02–102.71)	0.92
Elevated total bilirubin	3.52 (2.11–5.90)	<0.001	8.24 (2.76–24.63)	<0.001
Elevated lactate dehydrogenase	6.69 (1.57–28.46)	0.01	3.03 (0.13–73.13)	0.50

* The never-smoking status was used as a reference for the comparison of former, current, and unknown smoking status; CI: Confidence Interval.

## Data Availability

The data presented in this study are available on request from the corresponding author. The data are not publicly available due to Houston Methodist Research Institute Policies and Procedures related to Human Subject Research and Research Protections Good Clinical Practice procedures.

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
