# Peer review of "Carboxyhemoglobinemia in Critically Ill Coronavirus Disease 2019 Patients"

_jcm, 2021, doi:10.3390/jcm10122731_

Round 1

Reviewer 1 Report

Summary:

The authors report carboxyhemoglobin (Co-Hb) and methemoglobin-levels (Met-Hb) in their COVID19 patient cohort.

They report that the discrepancy between blood gas analyses and pulse oximetry was the reason why they further evaluated CO-Hb and Met-Hb in their patients, is it possible to provide data concerning on the number of patients who had false readings leading to reduced oxygen supply?

Is it correct that in the table normal COHb concentrations are <2.0% in non–smokers and >5% in current smokers and that still smokers the incidence of CO-Hb above >5% was higher than the level of >2% in non-smokers?

Methods:

It is not clear to me how the 431 patients with “hypoxia” were chosen out of the cohort of 5708 patients with COVID-19. How do you define hypoxia in your cohort? Please add this information in the method section.

Do all the tables and the statistical analysis refer to the first blood gas analysis at admission to the hospital or to the ICU? At which day of the disease course have these analyses been performed?

Is it possible to provide a figure on the timely course?

How many blood analyses have been assessed in each patient and at which time of the disease course have they been performed? Which measurement device has been used and on which pulse oximetry have the difference been detected?

Have patients out of this cohort been intubated and if yes, which medication has been used? What is the reason that you cannot add the information on mechanical ventilation of the patients?

Is there any information on the use of nitric oxide or the use of local anesthetics (such as used in some hospitals for propofol anesthesia). Is there any data concerning the use of nitric oxide?

Is HTN arterial hypertension and DM diabetes mellitus? Please define the abbreviations in the text

Venous, capillary and arterial measurements of hematocrit and hemoglobin differ significantly, which measurement have you relied on?

Results

Line 96: “Hypoxia” see comment above

In the table 1 with baseline-characteristics, the line showing sex-differences and the one with the differences concerning smokers is empty in the pdf file provided to me.

Likewise in table 2 the line smoker and smoker_never are empty.

As is published by Scholkmann et al (JCM, PMID 333 75707), there seems to be a timely variation between the levels of methemoglobin, which are increasing during the COVID19 disease course, could you add this information into your article?

Is it possible to provide the mean and median values as well as the standard deviation of methemoglobin-and carboxyhemoglobin-levels as measured?

Discussion:

Line 133:reference for your statement?

Could you refer to comparable viral or bacterial pulmonary diseases where methemoglobin-levels have been analysed?

Treatment options are not discussed, have there been any consequence taken out of the elevated methemoglobin-levels you measured? What would be your suggestion?

Author Response

Summary:

Point 1: The authors report carboxyhemoglobin (Co-Hb) and methemoglobin-levels (Met-Hb) in their COVID19 patient cohort.

Response 1: In our study, we are reporting only Carboxyhemoglobin (CO-Hb) in COVID-19 patients. Unfortunately, we do not have data on methemoglobin levels to report for this manuscript.

Point 2: They report that the discrepancy between blood gas analyses and pulse oximetry was the reason why they further evaluated CO-Hb and Met-Hb in their patients, is it possible to provide data concerning the number of patients who had false readings leading to reduced oxygen supply?

Response 2: Initially, these findings were observed by our critical care physicians taking care of Critically Covid-19 patients. Our study was not able to demonstrate any uniform results on the discrepancy between Oxygen saturation measured by pulse oximetry (SpO2) and arterial blood gas  (SaO2). We cannot provide accurate data on this information because arterial blood gase analysis was not performed simultaneously on many patients.

Pont 3: Is it correct that in the table normal COHb concentrations are <2.0% in non–smokers and >5% in current smokers and that still smokers the incidence of CO-Hb above >5% was higher than the level of >2% in non-smokers?

Response 3: Yes, the table is correct. Our laboratory department's at Houston Methodist Hospital follows the below reference range for CoHb level, which are as follows:

Carboxyhemoglobin

Comment: Reference Ranges: Carboxyhemoglobin
Non smoker:   0.0 - 2.0%
Smoker:       2.1 - 5.0%
Heavy smoker: 5.1 - 9%

Methods:

Point 1: It is not clear to me how the 431 patients with "hypoxia" were chosen out of the cohort of 5708 patients with COVID-19. How do you define hypoxia in your cohort? Please add this information in the method section.

Response 1: Hypoxia was defined as oxygen saturation measured by pulse oximetry (Spo2) < 94%. 

At our institute, the criteria for admission for the COVID-19 patients was hypoxia < 94% who requires virus-directed therapy (during the duration of the study). All 5708 patients with COVID-19 admitted at the Houston methodist hospital system had had hypoxia. However, only 431 patients were randomly assessed for COHb levels at any point during their disease course. (Response in methods)

Point 2: Do all the tables and the statistical analysis refer to the first blood gas analysis at admission to the hospital or to the ICU? At which day of the disease course have these analyses been performed?

Response 2: No.

The blood gas analysis and carboxyhemoglobin levels were checked randomly at any point during the disease course. ( Response in methods)

Pont 3: Is it possible to provide a figure on the timely course?

Response 3: We provide the figure on the mean COHb level of each patient with elevated and normal CoHb levels.  (Response in the result)

The mean initial COHb was 1.51±1.22%.

There was a decreasing trend with COHb levels in an abnormal/elevated COHb group throughout measurements (P-trend =0.01).

 However, the trend in the COHb level remained the same for the normal COHb  group (P-Trend >0.05

Point 4: How many blood analyses have been assessed in each patient and at which time of the disease course have they been performed? Which measurement device has been used and on which pulse oximetry have the difference been detected?

Response 4: Among 431 patients, 244 (56.6%) had one blood analysis, 187 (43.4%), had more than one blood analysis, and most patients had COHb level checked on a median hospital day of 2.5 days. {Inter quartile range (IQR): 0.1-7.6}.   (Response in the result)

We use a Co-Oximeter (ABL80 FLEX CO-OX blood gas analyzer) to check COHb levels at our institute. In addition, that it also provides the value of blood gas, electrolytes, glucose. An adult pulse oximeter adhesive sensor to check oxygen saturation was used in the patients of this study. (Response added in methods)

Point 5: Have patients out of this cohort been intubated and if yes, which medication has been used? What is the reason that you cannot add the information on mechanical ventilation of the patients?

Response 5: Out of 431 patients, 215 (49.88%) patients were admitted to intensive care units or inter-medicate care units.

149 out of 431 (34.57%) patients required intubation and mechanical ventilation. For the intubation medications, 137  (91%) patients received propofol, and rocuronium and  12 (8.05%) patients received propofol and succinylcholine for rapid sequence intubation.

(Response in the result)

Point 6: Is there any information on the use of nitric oxide or the use of local anesthetics (such as used in some hospitals for propofol anesthesia). Is there any data concerning the use of nitric oxide?

Response 6: All patients with elevated COHb levels did not require inhaled nitric oxide.

Point 7: Is HTN arterial hypertension and D.M. diabetes mellitus? Please define the abbreviations in the text

Response 7: HTN is essential hypertension and D.M. is diabetes mellitus. (Corrected in the Table 1)

Point 8:  Venous, capillary and arterial measurements of hematocrit and hemoglobin differ significantly, which measurement have you relied on?

Response 8: Hematocrit and haemoglobin levels were measured on venous blood as a complete blood count.

 Results

Point 1: Line 96: "Hypoxia" see comment above

Response 1: Hypoxia was defined as Spo2 < 94%. 

Point 2:  In the table 1 with baseline-characteristics, the line showing sex-differences and the one with the differences concerning smokers is empty in the pdf file provided to me.

Response 2: The table was fixed and re-arranged to read easily

Point 3: Likewise in table 2 the line smoker and smoker_never are empty.

Response 3: Table 2 was also corrected.

Point 4: As is published by Scholkmann et al (JCM, PMID 333 75707), there seems to be a timely variation between the levels of methemoglobin, which are increasing during the COVID19 disease course, could you add this information into your article?

Response 4: The article's citation was added—reference 11. However, we cannot comment on methemoglobin levels as we do not have data available on this variable.  

Point 5: Is it possible to provide the mean and median values as well as the standard deviation of methemoglobin-and carboxyhemoglobin-levels as measured?

Response 5: Yes. The mean value of COHb levels is added in the manuscript and above figures. 

Discussion:

Point 1: Line 133: reference for your statement?

Response 1: Done

Point 2: Could you refer to comparable viral or bacterial pulmonary diseases where methemoglobin-levels have been analysed?

Response 2: We do not have any information on methemoglobin data. Therefore, we are not able to add any comments on this point.

Point 3: Treatment options are not discussed, have there been any consequence taken out of the elevated methemoglobin-levels you measured? What would be your suggestion?

Response 3: We have added treatment interventions in the discussion part as follows.

Apart from hyperbaric or normobaric oxygen therapy and supportive care, treatment options for carboxyhemoglobinemia are limited. There is interest in hydroxocobalamin and ascorbic acid as a potential therapy given evidence conversion of COHb to carbon dioxide in rat models. Of note, hydroxocobalamin may result in falsely lower COHb levels. (Reference 1 & 2)

  1. Roderique JD, Josef CS, Newcomb AH, Reynolds PS, Somera LG, Spiess BD. Preclinical evaluation of injectable reduced hydroxocobalamin as an antidote to acute carbon monoxide poisoning. J Trauma Acute Care Surg. 2015;79(4 Suppl 2):S116-S120. doi:10.1097/TA.0000000000000740
  2. Pace R, Bon Homme M, Hoffman RS, Lugassy D. Effects of hydroxocobalamin on carboxyhemoglobin measured under physiologic and pathologic conditions. Clin Toxicol. 2014;52(7):647-650. doi:10.3109/15563650.2014.939659

Reviewer 2 Report

Elevated concentration of blood carboxhemoglobin (COHb) was previously observed in in few cases of critically ill COVID-19 patients, as also very recently reviewed by Scholkmann et al (J. Clin. Med. 2021, 10, 50).   

The Authors of the present manuscript for the first time have investigated the incidence of elevated carboxhemoglobinaemia on a large group of COVID-19 patients affected by severe hypoxia (n=431). COHb was over 5% in 27% of these patients. Considering that COHb limits the oxygen delivery, and standard two wavelength pulse oximetry is inaccurate and unreliable in the case of elevated COHb, these Authors wisely recommend checking blood COHb level routinely in critically ill COVID-19 patients.  

The results of the present study are extremely interesting and valuable to cope the necessity of a continuous update of COVID-19 guidelines. The data are clearly presented, and the statistics support consistently their conclusions.  

Scholkmann et al (J. Clin. Med. 2021, 10, 50) in their review article concluded that either methemoglobin (MetHb) or COHb, supposed to be elevated in COVID-19 patients, should be checked routinely in order to avoid misinterpretation of fingertip pulse oximetry readings. Therefore, it would be tremendously crucial/important and beneficial if the Authors could provide the readers also with metHb data.  

Nevertheless, independently on the availability of the metHb data, the Authors in their discussion might mention also the possibility to measure COHb, MetHb, and total hemoglobin blood concentration by non-invasive pulse CO-oximeters. This relatively low-cost point of care instrument, commercially available since 2005, can shorten the time to diagnosis and treatment of COVID-19 patients.  

Minor 

Line 63. >5% in current. Please clarify.  

Data of COHb repeated measurements during the time course of the disease might be included, if available. These data could be of great interest.

References: the review by Scholkmann et al (J. Clin. Med. 2021, 10, 50) should be quoted. 

Author Response

Response to Reviewer 2 Comments

Dear Editor/reviewer,

We are very thankful for the opportunity and for allowing us to revise our paper. This constructive advice has substantially improved the paper making it more transparent and informative.

Here are our detailed and line-by-line responses to the reviewer's comments.

We have made all changes with track changes in the revised manuscript.

Sincerely,

Hina Faisal, M.D., MRCS

Corresponding author & Principal investigator

=====================================================================

Point 1: Elevated concentration of blood carboxhemoglobin (COHb) was previously observed in in few cases of critically ill COVID-19 patients, as also very recently reviewed by Scholkmann et al (J. Clin. Med. 2021, 10, 50).   

The Authors of the present manuscript for the first time have investigated the incidence of elevated carboxhemoglobinaemia on a large group of COVID-19 patients affected by severe hypoxia (n=431). COHb was over 5% in 27% of these patients. Considering that COHb limits the oxygen delivery, and standard two wavelength pulse oximetry is inaccurate and unreliable in the case of elevated COHb, these Authors wisely recommend checking blood COHb level routinely in critically ill COVID-19 patients.  

 Response 1: We appreciate the positive comments

Point 2: The results of the present study are extremely interesting and valuable to cope the necessity of a continuous update of COVID-19 guidelines. The data are clearly presented, and the statistics support consistently their conclusions.  

Response 2: Appreciated

Point 3: Scholkmann et al (J. Clin. Med. 2021, 10, 50) in their review article concluded that either methemoglobin (MetHb) or COHb, supposed to be elevated in COVID-19 patients, should be checked routinely in order to avoid misinterpretation of fingertip pulse oximetry readings. Therefore, it would be tremendously crucial/important and beneficial if the Authors could provide the readers also with metHb data.  

Nevertheless, independently on the availability of the metHb data, the Authors in their discussion might mention also the possibility to measure COHb, MetHb, and total hemoglobin blood concentration by non-invasive pulse CO-oximeters. This relatively low-cost point of care instrument, commercially available since 2005, can shorten the time to diagnosis and treatment of COVID-19 patients.  

Response 3: Unfortunately, we lack the information and data on methemoglobin levels in our data set. Therefore, we can not provide this information.

 Minor 

Point 4: Line 63. >5% in current. Please clarify.  

Response 4: The information corrected

Point 5:  Data of COHb repeated measurements during the time course of the disease might be included, if available. These data could be of great interest.

Response 5:

Here, we provide the figure on the mean CoHb level of each patient with high and normal CoHb levels.  (Response in the result)

The mean initial COHb was 1.51±1.22%.

There was a decreasing trend with COHb levels in an abnormal COHb group throughout measurements (P-trend =0.01).

 However, the trend in the COHBlevel remained the same for the normal COHb  group (P-Trend >0.05

Point 6: References: the review by Scholkmann et al (J. Clin. Med. 2021, 10, 50) should be quoted. 

Response 6: We quoted and referenced it.

Reviewer 3 Report

This retrospective study presents the significant result that 26.7% of 431 critically ill COVID-19 patients had elevated carboxyhemoglobin levels.

I have only a few minor suggestions for the authors.

In the last sentence of page 5 of the manuscript under the Discussion section, there is a typo in "...excluding patients with sepsis and anemia, to see if COVID-19 infection is per se casing the COHb spike."  "casing" should be "causing".

In the second paragraph of page 2, the authors mentioned that "elevated COHb levels may continue to increase to toxic levels and turn deadly".  An increase of COHb level from 0.9% to 3.2% as reported in the manuscript certainly will not turn deadly.  Smokers routinely have COHb level at 5 to 10%.

The authors mentioned "critically ill" throughout the Introduction section and the Discussion section but not  in the Methods section.  In the Results section, the authors used the phrase "431 patients with hypoxia".  May I assume those selected patients were critically ill?

I basically accepted the paper submission.  Other than the typo, the rest is up to the authors to decide whether any modification is necessary. I do not need to review  the manuscript again.

Author Response

Response to Reviewer 3 Comments

Dear Editor/reviewer,

We are very thankful for the opportunity and for allowing us to revise our paper. This constructive advice has substantially improved the paper making it more transparent and informative.

Here are our detailed and line-by-line responses to the reviewer's comments.

We have made all changes with track changes in the revised manuscript.

Sincerely,

Hina Faisal, M.D., MRCS

Corresponding author & Principal investigator

=====================================================================

This retrospective study presents the significant result that 26.7% of 431 critically ill COVID-19 patients had elevated carboxyhemoglobin levels.

I have only a few minor suggestions for the authors.

Point 1: In the last sentence of page 5 of the manuscript under the Discussion section, there is a typo in "...excluding patients with sepsis and anemia, to see if COVID-19 infection is per se casing the COHb spike."  "casing" should be "causing".

Response1: We corrected it.

Point 2: In the second paragraph of page 2, the authors mentioned that "elevated COHb levels may continue to increase to toxic levels and turn deadly".  An increase of COHb level from 0.9% to 3.2% as reported in the manuscript certainly will not turn deadly.  Smokers routinely have COHb levels at 5 to 10%.

Response 2: Corrected it

Point 3: The authors mentioned "critically ill" throughout the Introduction section and the Discussion section but not  in the Methods section.  In the Results section, the authors used the phrase "431 patients with hypoxia".  May I assume those selected patients were critically ill?

Resonse 3: Yes. We have explained it more in the revised manuscript per other reviewer's and your's comments.

I basically accepted the paper submission.  Other than the typo, the rest is up to the authors to decide whether any modification is necessary. I do not need to review the manuscript again.

We appreciate a lot for accepting our paper for publication.

Round 2

Reviewer 1 Report

Thank you for revising your article

There are some minor comments:

Thank you for adding the name of the ABL-device, could you also add the provider/company?

Is it correct that you do not have the data of met-Hb available? I think that inflamed airways produce NO and CO which then lead to increased methemoglobin and carboxyhemoglobin-levels alike. That is why I asked about the Met-Hb-data.

I would suggest to rather refer to “intermediate care unit” instead of “intermedicate care unit”

Table 3: In the line  Smoker never, there occurs only a ref sign in my version

Thank you for provision of the time course. Do I read it correctly that some patients had more than 50 measurements of CO-Hb?

Author Response

Point 1: Name of the ABL-device, could you also add the provider/company?

Response 1: ABL80 FLEX blood gas analyzer (Company: Radiometer; Provider: Houston Methodist Hospital).

Point 2: Is it correct that you do not have the data of met-Hb available? I think that inflamed airways produce NO and CO which then lead to increased methemoglobin and carboxyhemoglobin-levels alike. That is why I asked about the Met-Hb-data.

Response 2: Correct. Currently, we do not have data available on the Met-Hb level. I agree with your explanation. We will try to collect the data on Met-Hb in the future for future studies.

Point 3: I would suggest to rather refer to "intermediate care unit" instead of "intermedicate care unit."

Response: It is the "Intermediate care unit."

Point 3: Table 3: In the line  Smoker never, there occurs only a ref sign in my version.

Response 3: Smoking status had 4 categories, so the first category (never smoker) was as a reference group. All others (former, current, and unknown) compared with it. For other variables in table 3, all were binary variables. For example, for pre-existing anemias, patients with pre-existing anemias (yes) compared with patients without pre-existing anemias. We have placed a unique sign * with the explanation under the table.

Point 4: Do I read it correctly that some patients had more than 50 measurements of CO-Hb?

Response 4: Yes, you read it correctly. The maximum repetition for measurements was 65.

  Min. 1st Qu.  Median    Mean 3rd Qu.    Max.

  1.000   1.000   1.000   5.701   4.000  65.000